# Association of body mass index with health care expenditures in the United States by age and sex

**Zachary J. Ward**[1]*, **Sara N. Bleich**[2], **Michael W. Long**[3], **Steven L. Gortmaker**[4]

**1** Center for Health Decision Science, Harvard T. H. Chan School of Public Health, Boston, MA, United States of America, **2** Department of Health Policy and Management, Harvard T. H. Chan School of Public Health, Boston, MA, United States of America, **3** Department of Prevention and Community Health, Milken Institute School of Public Health, George Washington University, Washington, DC, United States of America, **4** Department of Social and Behavioral Sciences, Harvard T. H. Chan School of Public Health, Boston, MA, United States of America

* zward@hsph.harvard.edu

## Abstract

### Background

Estimates of health care costs associated with excess weight are needed to inform the development of cost-effective obesity prevention efforts. However, commonly used cost estimates are not sensitive to changes in weight across the entire body mass index (BMI) distribution as they are often based on discrete BMI categories.

### Methods

We estimated continuous BMI-related health care expenditures using data from the Medical Expenditure Panel Survey (MEPS) 2011–2016 for 175,726 respondents. We adjusted BMI for self-report bias using data from the National Health and Nutrition Examination Survey (NHANES) 2011–2016, and controlled for potential confounding between BMI and medical expenditures using a two-part model. Costs are reported in $US 2019.

### Results

We found a J-shaped curve of medical expenditures by BMI, with higher costs for females and the lowest expenditures occurring at a BMI of 20.5 for adult females and 23.5 for adult males. Over 30 units of BMI, each one-unit BMI increase was associated with an additional cost of $253 (95% CI $167-$347) per person. Among adults, obesity was associated with $1,861 (95% CI $1,656-$2,053) excess annual medical costs per person, accounting for $172.74 billion (95% CI $153.70-$190.61) of annual expenditures. Severe obesity was associated with excess costs of $3,097 (95% CI $2,777-$3,413) per adult. Among children, obesity was associated with $116 (95% CI $14-$201) excess costs per person and $1.32 billion (95% CI $0.16-$2.29) of medical spending, with severe obesity associated with $310 (95% CI $124-$474) excess costs per child.

**Data Availability Statement:** Data from the Medical Expenditure Panel Survey (MEPS) are available at: https://www.meps.ahrq.gov/mepsweb/, and data from the National Health and Nutrition

Examination Survey (NHANES) are available at:
http://www.cdc.gov/nchs/nhanes.htm. Estimates of
age/sex/BMI-specific annual medical costs are
available at: https://doi.org/10.7910/DVN/872OW1.

**Funding:** All of the authors received support from
The JPB Foundation (Grant No. 1085). ZJW, SNB,
and SLG were supported by the National Institutes
of Health (Grant No. R01HL146625). SNB and SLG
were supported by the Centers for Disease Control
and Prevention (CDC) (Grant No. U48DP006376).
This work is solely the responsibility of the authors
and does not represent official views of the CDC or
other agencies. The funders had no role in study
design, data collection and analysis, decision to
publish, or preparation of the manuscript.

**Competing interests:** The authors have declared
that no competing interests exist.

## Conclusions

Higher health care costs are associated with excess body weight across a broad range of
ages and BMI levels, and are especially high for people with severe obesity. These findings
highlight the importance of promoting a healthy weight for the entire population while also
targeting efforts to prevent extreme weight gain over the life course.

## Introduction

Seven out of ten adults and three out of ten children in the United States currently have over-
weight or obesity [1, 2], and the prevalence continues to rise, with half of US adults projected
to have obesity by 2030 [3], and nearly 60% of today's children predicted to have obesity by
age 35 [4]. Excess body weight is associated with a wide array of comorbidities and premature
mortality [5], and higher health care costs [6–13], which are expected to increase as population
body mass index (BMI) continues to rise in the United States [13].

Accurate estimates of excess weight-related health care costs are necessary to evaluate the
cost-effectiveness of policies and programs aimed at helping to reverse the obesity epidemic
and promote a healthy weight across a range of ages and BMI levels [14, 15]. However, com-
monly used cost estimates are not sensitive to changes in weight across the entire BMI distri-
bution as they are often based on discrete categories, such as binary classification (obesity vs
non-obesity) or BMI categories (e.g. moderate vs severe obesity) [12, 13]. Using discrete cate-
gories likely underestimates the health care cost impact of changes in population BMI, as only
changes in weight that cross specific category thresholds are accounted for, therefore ignoring
changes within categories at all other parts of the BMI distribution. In contrast, estimating
continuous BMI-related costs provides a more accurate and flexible approach as it reflects the
entire BMI distribution and does not rely on specific category thresholds.

To address key gaps in the literature, in this paper we estimate continuous BMI-specific
medical costs by age and sex, and provide updated estimates of the medical costs attributable
to obesity using recent data.

## Methods

### Data

We used publicly available, de-identified data from the Medical Expenditure Panel Survey
(MEPS) 2011–2016. We harmonized variable definitions across years and adjusted total expen-
ditures to $US 2019 using the Personal Consumption Expenditures (Health) index [16]. After
excluding pregnant women and respondents with missing variables of interest, our pooled
dataset contained 175,726 respondents– 139,143 adults (aged 20 and older) and 36,583 chil-
dren (aged 6 to 19). BMI was not available for children younger than 6. See S1 File, Section 1
for details on dataset harmonization, exclusion criteria, and respondent characteristics.

### Adjustment for self-report bias

We adjusted reported BMI in MEPS to correct for self-report bias that leads to underestimates
of obesity prevalence [17, 18]. We used a semi-parametric method [3] to adjust the distribution
of self-reported BMI to match nationally-representative, measured data from the National
Health and Nutrition Examination Survey (NHANES). Specifically, we used cubic splines to
estimate the magnitude of self-report bias by BMI quantile and adjust the self-reported BMI in

MEPS by age group. Using this approach our sex-specific distributions of adjusted BMI were statistically similar to NHANES (p>0.05). See S1 File, Section 2 for details.

## Expenditure standardization

To adjust for potential confounding of the relationship between BMI and medical expenditures, we standardized respondents' expenditures to be representative of a synthetic, 'average' population, thus controlling for the effects of other salient factors (e.g. smoking, insurance coverage, etc.). Using a well-established approach [7, 9, 11], we fit a two-part regression model to predict medical expenditures. The first part of the model fit a logistic regression to predict the probability of non-zero expenditures. The second part of the model fit a linear regression of the log expenditure given positive expenditure. The two parts were then multiplied together to yield the full model.

Similar to previous analyses [9, 11], we controlled for the following variables: BMI (continuous), year (continuous), geographical region (Northeast, Midwest, South, West), age (continuous), sex, race/ethnicity (White, Black, Hispanic, American Indian/Alaska Native, Asian/Native Hawaiian/Pacific Islander, multiple races), marital status (married, widowed, divorced, separated, never married), education (less than high school, some high school, GED or high school diploma, some college, college graduate, graduate school, unknown), smoking status (yes/no), poverty level (continuous), and insurance coverage (private, TRICARE, Medicare, Medicaid, other public A, other public B, none). We fit separate models for children and adults, and did not control for education or marital status when adjusting expenditures for children. Continuous variables were modeled as cubic polynomials for greater flexibility, and were standardized to reduce multicollinearity and improve numerical stability [19].

We used ridge regression [20, 21] to help guard against over-fitting to extreme values that can occur given the highly skewed nature of health expenditures [22]. Using the fitted two-part model we then adjusted each respondent's probability and level of total expenditure to be representative of a standardized individual. We adjusted for all variables except BMI (and age when fitting bivariate models–see below), thus controlling expenditures for other salient factors. See S1 File, Section 3 for details.

## Predicted expenditures by BMI

We then used generalized additive models (GAMs) [23] to estimate the relationship between log BMI and the log of the standardized expenditures using the same two-part model described above. We fit two different types of models: a univariate model (cubic smoothing splines) with log BMI as a continuous predictor, and a bivariate model [24] with both age and log BMI as continuous predictors to capture the interaction between age and BMI (see S1 File, Section 4.1 for details). We predicted adult expenditures for BMI values between 10 and 80. For children we predicted expenditures for BMI z-scores between -3 and +3. Although we fitted our model using BMI for both children and adults, we present the univariate predicted expenditures for children in terms of BMI z-score for visual ease, as the overweight and obesity thresholds are defined by BMI z-score for children. With these predicted expenditures we also estimated costs by binary obesity status: non-obesity vs obesity, and by BMI category: underweight, normal weight, overweight, moderate obesity, severe obesity (see S1 File, Section 4.2 for category definitions).

To estimate population-level excess costs we assumed that all respondents would instead follow the BMI distribution observed in the reference category (i.e. weight category with lowest costs) and re-estimated the total costs. Excess costs were then calculated as the difference between the current predicted costs and the predicted costs for the reference weight

population. We scaled the per-person excess costs to the population-level using BMI category prevalence estimates from NHANES 2011–2016 and 2019 population estimates of the civilian, non-institutionalized population [25].

We also estimated how categorical expenditures change with age by fitting smooth splines to the predicted costs within each BMI category by age. Because age is top-coded at 85 years old in MEPS, we predicted costs from ages 6 to 85.

### Model uncertainty

To estimate confidence intervals for all results we bootstrapped the MEPS dataset 1,000 times, taking into account the complex survey structure, and re-estimated all models described above. We calculated 95% confidence intervals (CIs) as the 2.5 and 97.5 percentiles of the bootstrapped results (see S1 File, Section 5 for details). All analyses were performed in R (version 3.6.1).

## Results

### Adults

We found a J-shaped curve of medical expenditures by BMI (Fig 1A), with higher costs in general for females and the lowest expenditures occurring at a BMI of 20.5 for females and 23.5 for males. Above a BMI of 30, predicted costs continued to increase linearly, with each one-unit increase in BMI associated with an additional cost of $253 (95% CI $167-$347) per person on average. Obesity was associated with $1,861 (95% CI $1,656-$2,053) excess annual medical costs per person, accounting for nearly $173 billion (95% CI $153.70-$190.61) of annual spending in the US (Table 1). Most of these costs are from individuals with severe obesity, who have excess annual costs of over $3,000 (95% CI $2,777-$3,413). We also found that having overweight is associated with over $600 per person in excess costs (95% CI $503-$756), contributing to $50 billion (95% CI $40.64-$61.12) in medical spending per year.

### Children

We found a very shallow J-shaped curve of medical expenditures by BMI z-score for boys, meaning that we observed higher expenditures for boys with the lowest BMI z-scores and only a slight increase in expenditures as z-scores increased from the lowest expenditure level. Expenditures for girls were lower and did not exhibit increased costs at low BMI z-scores (Fig 1B). For both boys and girls, we found that medical expenditures only increased substantially over the 99[th] percentile of BMI. Among children, obesity is associated with $116 (95% CI $14-$201) excess annual medical costs per person and $1.32 billion (95% CI $0.16-$2.29) of medical spending (Table 1). By BMI category we find increased costs for children with severe obesity of over $300 per person a year (95% CI $124-$474).

### Age-specific

For predictions by BMI and age (Fig 1C) we found a similar J-shaped relationship of expenditures by BMI at all ages, with increasing expenditures by age. The highest predicted expenditures are for individuals with severe obesity between 60–70 years of age.

Our predictions of expenditure by obesity status and BMI category revealed increasing costs in all groups by age, with differential cost increases for obesity, especially for severe obesity (Fig 2). We found that severe obesity is associated with increased costs at all ages. BMI-specific medical costs by age and sex are available in a public repository (https://doi.org/10.7910/DVN/872OW1).

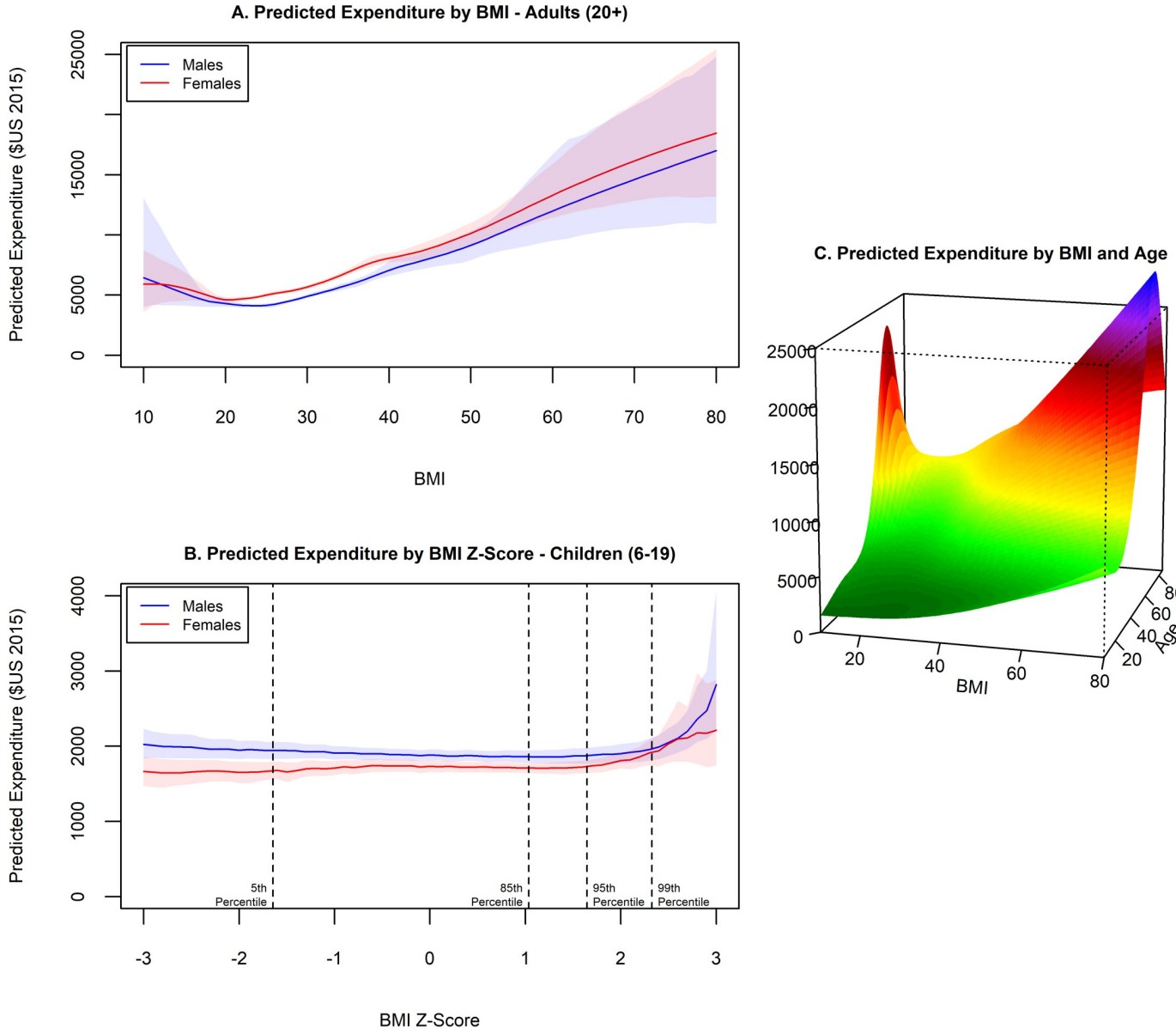

**Fig 1. Estimated BMI-related medical expenditures, children and adults.** Estimated expenditures are controlled for potential confounding variables. Shaded areas represent 95% confidence intervals.

## Discussion

Using a continuous costing approach, we provide updated national and individual-level estimates of the annual direct health care costs associated with excess weight for children and adults in the US. We found a J-shaped curved of medical expenditures by BMI for adults, consistent with the epidemiologic data on BMI-related mortality [5]. The lowest predicted medical costs for adults occurred at a BMI between 20 and 24 for all ages. We find that among adults, obesity is associated with over $1,800 excess annual medical costs per person, accounting for over $170 billion of annual spending in the US. This figure rises to over $200 billion if excess costs from overweight (over $600 per person) are included, highlighting the large economic impact of overweight and obesity in the US.

**Table 1. Total and excess annual medical expenditures by BMI category ($US 2019).**

| Obesity Status (BMI range) | Total Cost Per Person[a] (95% CI) | Excess Cost Per Person[b] (95% CI) | Excess Cost (Billions)–Population-level[c] (95% CI) |
|---|---|---|---|
| **Children (6–19)[d]** | | | |
| Non-Obesity (BMI < 95%ile) | 1,871 (1,826–1,918) | Reference | Reference |
| Obesity (BMI ≥ 95%ile) | 1,987 (1,893–2,072) | 116 (14–201) | 1.32 (0.16–2.29) |
| *BMI Category* | | | |
| Underweight (BMI < 5%ile) | 1,913 (1,806–2,039) | 41 (-59-161) | 0.09 (-0.13–0.35) |
| Normal weight (5%ile ≤ BMI < 85%ile) | 1,873 (1,821–1,925) | Reference | Reference |
| Overweight (85%ile ≤ BMI < 95%ile) | 1,852 (1,794–1,916) | -21 (-84-45) | -0.21 (-0.83–0.44) |
| Moderate Obesity (95%ile ≤ BMI < 120% x 95%ile) | 1,882 (1,803–1,957) | 9 (-83-96) | 0.07 (-0.61–0.70) |
| Severe Obesity (BMI ≥ 120% x 95%ile) | 2,183 (2,013–2,327) | 310 (124–474) | 1.27 (0.51–1.94) |
| **Adults (20+)** | | | |
| Non-Obesity (BMI < 30) | 4,525 (4,450–4,616) | Reference | Reference |
| Obesity (BMI ≥ 30) | 6,385 (6,221–6,558) | 1,861 (1,656–2,053) | 172.74 (153.70–190.61) |
| *BMI Category* | | | |
| Underweight (BMI < 18.5) | 4,419 (3,970–4,921) | 228 (-201-721) | 0.85 (-0.75–2.68) |
| Normal weight (18.5 ≤ BMI < 25) | 4,191 (4,092–4,306) | Reference | Reference |
| Overweight (25 ≤ BMI < 30) | 4,812 (4,716–4,936) | 621 (503–756) | 50.19 (40.64–61.12) |
| Moderate Obesity (30 ≤ BMI < 35) | 5,672 (5,548–5,808) | 1,480 (1,305–1,650) | 77.03 (67.91–85.83) |
| Severe Obesity (BMI ≥ 35) | 7,288 (7,002–7,594) | 3,097 (2,777–3,413) | 126.39 (113.35–139.29) |

[a] Mean of predicted costs for respondents in each BMI category, controlling for age, sex, and other covariates in the two-part model.

[b] Excess costs were estimated by assuming that all respondents would instead follow the BMI distribution observed in the reference category, then calculating the difference between the current predicted costs and the predicted costs for the reference weight population.

[c] Population-level costs were estimated by scaling the per-person excess costs using BMI category prevalence estimates from NHANES 2011–2016 and 2019 population estimates of the civilian, non-institutionalized population.

[d] %ile = percentile.

For children, we find that across most of the range of BMI z-scores there was no association with health care expenditures, with large increases in costs occurring only above the 99th percentile of BMI. Overall, we find that childhood obesity is associated with over $100 per child with obesity and over $1 billion of excess medical costs. Health care expenditures for children with severe obesity were increased by $300 per person per year. Although children's obesity costs are a relatively small contributor to excess medical spending (less than 1% of all obesity-related medical expenditures), preventing childhood obesity may help to avert future health care costs given that excess body weight during childhood is a strong predictor of excess weight during adulthood [4].

We find that obesity-related costs increase with age, starting around age 30. This is similar to findings by the Global Burden of Disease and Global BMI Mortality Collaboration that report increased relative risks of obesity-related morbidity and mortality starting at ages 25–29 and 35+, respectively [5, 26]. Thus, our findings of little excess cost at younger ages is consistent with the epidemiological evidence that obesity-related disease mostly occurs later in life.

However, even at younger ages we do find increased costs associated with severe obesity, highlighting the importance of preventing extreme weight gain at all ages. The high costs at higher levels of BMI is especially concerning given that the prevalence of severe obesity among adults is projected to increase further and become the most common BMI category for some subgroups [3].

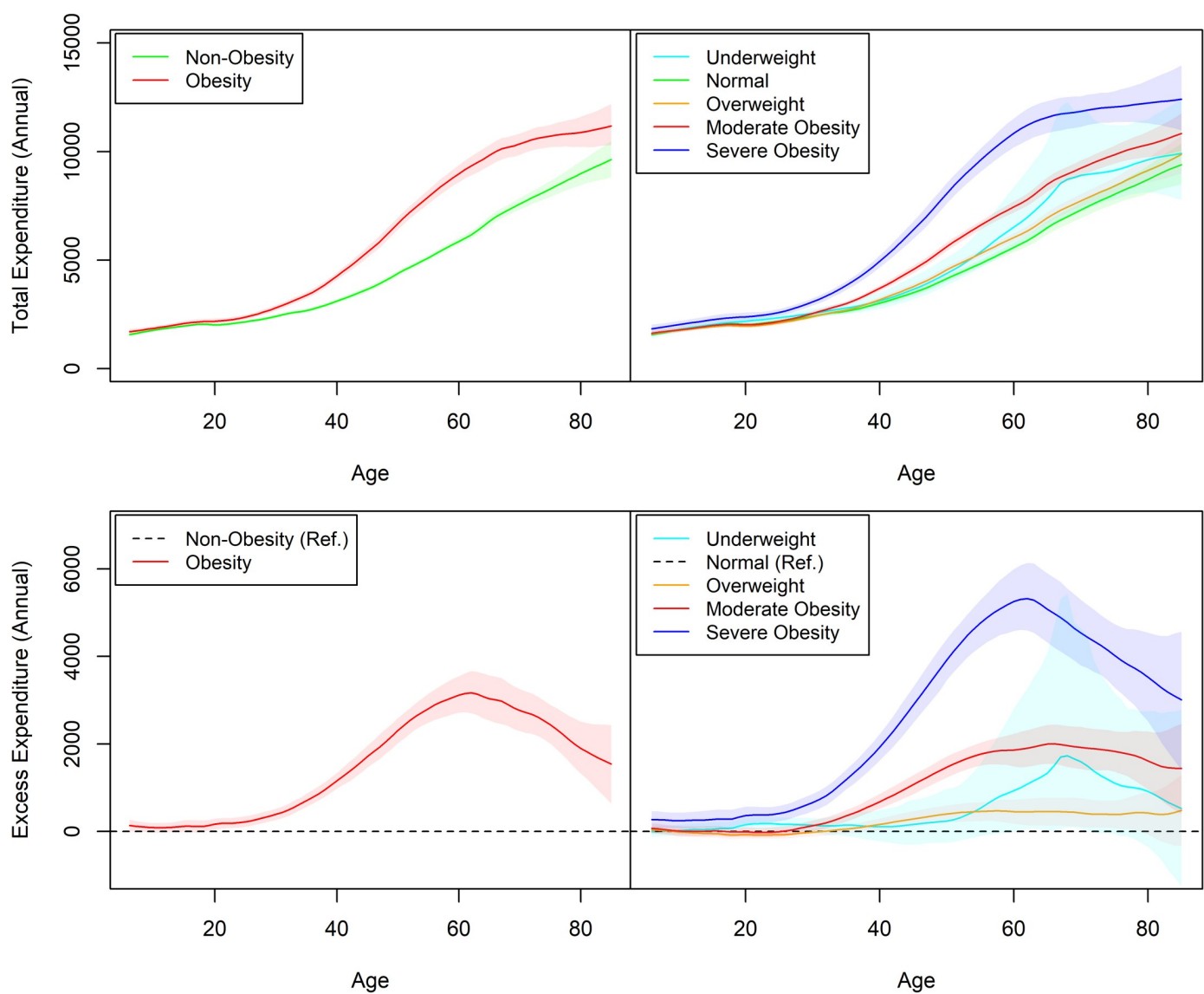

**Fig 2. Estimated age-specific medical expenditures by BMI category.** Estimated expenditures are controlled for potential confounding variables. Shaded areas represent 95% confidence intervals.

Our results suggest that obesity-related excess costs increase with age until about age 65, at which point the gap between obesity and non-obesity begins to narrow. This is partly due to increasing costs among those with normal weight as a result of ageing. However, it is also driven by a flattening out of costs in the severe obesity group; the costs for this group plateau while they continue to increase for all other BMI categories over this age range. This observed flattening may be due to selection bias (i.e. informative censoring) caused by higher mortality among individuals with severe obesity. We see that excess costs peak at progressively later ages for moderate obesity and overweight compared to severe obesity, adding support to the idea of mortality-induced censoring. Models to estimate the cost-effectiveness of obesity interventions among adults thus need to take into account effects on mortality as well [14].

Our estimates of obesity-related excess medical costs are similar to but higher than previous estimates by Finkelstein [6] and Wang [11], but lower than estimates by Cawley [10] using an

instrumental variable (IV) approach (see S1 File, Section 6 for details of comparisons to previous estimates). Our overall results are also similar to a meta-analysis of 12 studies which estimated that obesity-attributable medical costs were $1,901 (95% CI $1,239-$2,582) per person in 2014 $US, accounting for $150 billion at the national level [12]. Consistent with previous findings [27], we found that overall per capita spending was higher for adult women than adult men, but lower for girls than boys. Also similar to past research [13], we found higher excess obesity-related costs for adult women, but no difference for boys and girls.

## Limitations

While we control for a broad range of covariates to estimate the relationship between BMI and medical expenditures, there may be residual confounding due to unobserved variables, such as physical activity. In addition, our estimates are based on the cross-sectional association between BMI and medical expenditures. Large-scale longitudinal data tracking changes in individual-level BMI and expenditures over time would help more firmly establish the causal relationship between BMI and medical costs.

Also, we pooled all MEPS data from 2011–2016 to improve the stability of our estimates, so we could not examine trends within this period. Lastly, we only considered direct medical costs of obesity in this study. Including the indirect costs of obesity, such as lost wages due to obesity-related illness or disability or loss of future earnings due to premature death [28], would provide more comprehensive estimates of the economic impact of obesity.

## Conclusions

We found that health care expenditures are higher for people with excess weight across a wide range of ages and BMI levels, with especially high costs for people with severe obesity. Obesity-related medical costs are higher for adult females, and increase with age for all adults, with the highest estimated costs occurring for 60–70 year olds. Although childhood obesity contributes a small proportion of total obesity-related medical costs, because excess weight in childhood is a strong predictor of adult obesity, policies to prevent excess weight gain at all ages are needed to mitigate the health and economic impact of the obesity epidemic, which accounts for over $170 billion in excess medical costs per year in the United States. These findings highlight the importance of promoting healthy weight across the entire BMI distribution, and provide policy makers and practitioners with more accurate estimates of the health care cost impact of excess weight by age, sex, and continuous BMI.

## Supporting information

**S1 File. Supplemental appendix.** Additional methodological details and results.
(PDF)

## Author Contributions

**Conceptualization:** Zachary J. Ward, Sara N. Bleich, Michael W. Long, Steven L. Gortmaker.

**Data curation:** Zachary J. Ward.

**Formal analysis:** Zachary J. Ward.

**Methodology:** Zachary J. Ward.

**Software:** Zachary J. Ward.

**Writing – original draft:** Zachary J. Ward.

**Writing – review & editing:** Zachary J. Ward, Sara N. Bleich, Michael W. Long, Steven L. Gortmaker.

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
