## [Decision Letter · Decision Letter 0]

18 Dec 2020

PONE-D-20-36222

Association of body mass index with health care expenditures in the United States by age and sex

PLOS ONE

Dear Dr. Ward,

Thank you for submitting your manuscript to PLOS ONE. After careful consideration, we feel that it has merit but does not fully meet PLOS ONE’s publication criteria as it currently stands. Therefore, we invite you to submit a revised version of the manuscript that addresses the points raised during the review process.

We look forward to receiving your revised manuscript.

Kind regards,

Robert Siegel

Academic Editor

PLOS ONE

Additional Editor Comments:

Please address all the reviewer comments and suggestions.

Journal Requirements:

3. You have not indicated whether ethical approval was waived or necessary for your study. We understand that the framework for ethical oversight requirements for studies of this type may differ depending on the setting and we would appreciate some further clarification regarding your research. Could you please provide further details on ethical oversight of your study. Please clarify whether or why your study is exempt from the need for approval.

Reviewers' comments:

Reviewer's Responses to Questions

**Comments to the Author**

1. Is the manuscript technically sound, and do the data support the conclusions?

Reviewer #1: Yes

Reviewer #2: Yes

2. Has the statistical analysis been performed appropriately and rigorously? 

Reviewer #1: Yes

Reviewer #2: Yes

3. Have the authors made all data underlying the findings in their manuscript fully available?

Reviewer #1: Yes

Reviewer #2: Yes

4. Is the manuscript presented in an intelligible fashion and written in standard English?

Reviewer #1: Yes

Reviewer #2: Yes

5. Review Comments to the Author

Reviewer #1: This is a very carefully constructed analysis of the cost of continuously, in addition to categorically, measured BMI considering both age and sex with which I can find no fault in method.

I have two suggestions for improvement in the paper. The first is to explain why you undertake all the corrections for self-report bias relative to NHANES rather than simply using NHANES. It seems like NHANES may not be sufficiently representative, but it is unclear as they cover the same time period.

The second in the development of the conclusions. The paper set out to “To address key gaps in the literature, in this paper we estimate continuous BMI-specific medical costs by age and sex, and provide updated estimates of the medical costs attributable to obesity using recent data.” Please link back to this in the conclusions—note the higher costs of females, the comparatively low cost of children, and the high costs of 60-70 year olds. Linking back the billions of dollars in excess costs as a motivation (including opportunity cost) as an immediate call to policy action could be emphasized.

Reviewer #2: I appreciate this very relevant and important topic.

The data and information as presented for the obesity-related costs by BMI units as well as by BMI percentile categories is very clear. There is limited data out there for pediatrics so greatly appreciate the inclusion of pediatrics from ages 6-19 as well as your ability to control for all the many variables listed and the use of separate models for children vs adults.

I did have a question about the use of z-scores in children vs the use of actual BMI values like in the adults as the use of z- scores in the literature has become somewhat controversial.

You may want to consider reviewing this article below from 2017 that is similar in nature (but nto the same) to your study.

The Additional Costs and Health Effects of a Patient Having Overweight or Obesity: A Computational Model

Saeideh Fallah-Fini1,2, Atif Adam1, Lawrence J. Cheskin1, Sarah M. Bartsch1, and Bruce Y. Lee

Obesity (2017) 25, 1809-1815. doi:10.1002/oby.21965

6. PLOS authors have the option to publish the peer review history of their article (what does this mean?). If published, this will include your full peer review and any attached files.

Reviewer #1: No

Reviewer #2: No

---

## [Author Response · Author response to Decision Letter 0]

22 Jan 2021

Additional Editor Comments:

We have revised the manuscript formatting and file naming to meet the style requirements. 

We used public use samples of de-identified datasets which are freely available online. We have amended the Methods section of the manuscript to clarify this point.

Line 24: “We used publicly available, de-identified data from the Medical Expenditure Panel Survey (MEPS) 2011-2016.”

3. You have not indicated whether ethical approval was waived or necessary for your study. We understand that the framework for ethical oversight requirements for studies of this type may differ depending on the setting and we would appreciate some further clarification regarding your research. Could you please provide further details on ethical oversight of your study. Please clarify whether or why your study is exempt from the need for approval.

This work does not involve human subjects, and is thus IRB exempt. We used public use samples of de-identified datasets which are freely available online. 

We have revised the cover letter to provide this information. We have also added our estimates of age/sex/BMI-specific medical expenditures to a data repository, which will be made publicly available if the analysis is accepted for publication.

Lines 132-133:

“BMI-specific medical costs by age and sex are available in a public repository (https://doi.org/10.7910/DVN/872OW1).”

Reviewers' comments:

Reviewer #1: 

This is a very carefully constructed analysis of the cost of continuously, in addition to categorically, measured BMI considering both age and sex with which I can find no fault in method.

Thank you for your supportive comments.

I have two suggestions for improvement in the paper. The first is to explain why you undertake all the corrections for self-report bias relative to NHANES rather than simply using NHANES. It seems like NHANES may not be sufficiently representative, but it is unclear as they cover the same time period.

Thank you for this clarifying question. NHANES is generally considered the gold-standard for nationally-representative, measured BMI. However, NHANES does not have information on medical costs – the outcome of interest in this analysis. Although MEPS does have individual-level data on both medical costs and BMI, the BMI data are self-reported, which is well-known to cause bias – generally underestimating BMI for adults, and often overestimating it for young children. Because NHANES and MEPS are both designed to be nationally-representative samples, we can use information on the BMI distribution from NHANES to adjust MEPS data for self-reporting bias, using the corrected individual-level BMI to more accurately estimate the relationship between BMI and medical expenditures.

The second in the development of the conclusions. The paper set out to “To address key gaps in the literature, in this paper we estimate continuous BMI-specific medical costs by age and sex, and provide updated estimates of the medical costs attributable to obesity using recent data.” Please link back to this in the conclusions—note the higher costs of females, the comparatively low cost of children, and the high costs of 60-70 year olds. Linking back the billions of dollars in excess costs as a motivation (including opportunity cost) as an immediate call to policy action could be emphasized.

Thank you for this suggestion. We have highlighted these findings in the conclusions.

Lines 194-203:

“We found that health care expenditures are higher for people with excess weight across a wide range of ages and BMI levels, with especially high costs for people with severe obesity. Obesity-related medical costs are higher for adult females, and increase with age for all adults, with the highest estimated costs occurring for 60-70 year olds. Although childhood obesity contributes a small proportion of total obesity-related medical costs, because excess weight in childhood is a strong predictor of adult obesity, policies to prevent excess weight gain at all ages are needed to mitigate the health and economic impact of the obesity epidemic, which accounts for over $170 billion in excess medical costs per year in the United States. These findings highlight the importance of promoting healthy weight across the entire BMI distribution, and provide policy makers and practitioners with more accurate estimates of the health care cost impact of excess weight by age, sex, and continuous BMI.”

Reviewer #2: 

I appreciate this very relevant and important topic. The data and information as presented for the obesity-related costs by BMI units as well as by BMI percentile categories is very clear. There is limited data out there for pediatrics so greatly appreciate the inclusion of pediatrics from ages 6-19 as well as your ability to control for all the many variables listed and the use of separate models for children vs adults.

Thank you for your supportive comments. We agree that data for children are often not available, and felt it was important to include them in this analysis

I did have a question about the use of z-scores in children vs the use of actual BMI values like in the adults as the use of z- scores in the literature has become somewhat controversial.

Thank you for the opportunity to clarify this point. We did use the actual BMI values when estimating the models, but use the corresponding z-scores to display the predicted expenditures in the figure. We have clarified this in the manuscript.

Lines 68-72:

“We predicted adult expenditures for BMI values between 10 and 80. For children we predicted expenditures for BMI z-scores between -3 and +3. Although we fitted our model using BMI for both children and adults, we present the univariate predicted expenditures for children in terms of BMI z-score for visual ease, as the overweight and obesity thresholds are defined by BMI z-score for children.”

You may want to consider reviewing this article below from 2017 that is similar in nature (but nto the same) to your study.

The Additional Costs and Health Effects of a Patient Having Overweight or Obesity: A Computational Model Saeideh Fallah-Fini1,2, Atif Adam1, Lawrence J. Cheskin1, Sarah M. Bartsch1, and Bruce Y. Lee Obesity (2017) 25, 1809-1815. doi:10.1002/oby.21965

Thank you for this suggestion. The referenced paper estimates incremental costs over an adult’s lifetime using a Markov model, with the direct medical costs of overweight/obesity vs normal weight one of the inputs to the model, based on data from MEPS. However it is not clear that the authors adjusted for self-report bias or potential confounding when analyzing the MEPS data. Our current analysis aims to provide such cost estimates which could be used in simulation models such as this.

---

## [Decision Letter · Decision Letter 1]

5 Feb 2021

Association of body mass index with health care expenditures in the United States by age and sex

PONE-D-20-36222R1

Dear Dr. Ward,

We’re pleased to inform you that your manuscript has been judged scientifically suitable for publication and will be formally accepted for publication once it meets all outstanding technical requirements.

Kind regards,

Robert Siegel

Academic Editor

PLOS ONE

Additional Editor Comments (optional):

You have a successfully addressed all the reviewer concerns

Reviewers' comments:

Reviewer's Responses to Questions

**Comments to the Author**

1. If the authors have adequately addressed your comments raised in a previous round of review and you feel that this manuscript is now acceptable for publication, you may indicate that here to bypass the “Comments to the Author” section, enter your conflict of interest statement in the “Confidential to Editor” section, and submit your "Accept" recommendation.

Reviewer #1: All comments have been addressed

2. Is the manuscript technically sound, and do the data support the conclusions?

Reviewer #1: Yes

3. Has the statistical analysis been performed appropriately and rigorously? 

Reviewer #1: Yes

4. Have the authors made all data underlying the findings in their manuscript fully available?

Reviewer #1: Yes

5. Is the manuscript presented in an intelligible fashion and written in standard English?

Reviewer #1: Yes

6. Review Comments to the Author

Reviewer #1: (No Response)

7. PLOS authors have the option to publish the peer review history of their article (what does this mean?). If published, this will include your full peer review and any attached files.

Reviewer #1: No

---

## [Editor Report · Acceptance letter]

2 Mar 2021

PONE-D-20-36222R1 

Association of body mass index with health care expenditures in the United States by age and sex 

Dear Dr. Ward:

I'm pleased to inform you that your manuscript has been deemed suitable for publication in PLOS ONE. Congratulations! Your manuscript is now with our production department. 

Kind regards, 

on behalf of

Dr. Robert Siegel 

Academic Editor

PLOS ONE